# Differentiation and Establishment of Dental Epithelial-Like Stem Cells Derived from Human ESCs and iPSCs

**DOI:** 10.3390/ijms21124384

**Published:** 2020-06-19

**Authors:** Gee-Hye Kim, Jihye Yang, Dae-Hyun Jeon, Ji-Hye Kim, Geun Young Chae, Mi Jang, Gene Lee

**Affiliations:** Laboratory of Molecular Genetics, Dental Research Institute, School of Dentistry, Seoul National University, Seoul 03080, Korea; haha38@snu.ac.kr (G.-H.K.); yangjh56@snu.ac.kr (J.Y.); daedae7@snu.ac.kr (D.-H.J.); yellowpengin@snu.ac.kr (J.-H.K.); cgy0627@snu.ac.kr (G.Y.C.)

**Keywords:** dental development, human embryonic stem cells, human induced pluripotent stem cells, Hertwig’s epithelial root sheath/epithelial rests of Malassez cells, epithelial–mesenchymal interaction, hard tissue remodeling

## Abstract

Tooth development and regeneration occur through reciprocal interactions between epithelial and ectodermal mesenchymal stem cells. However, the current studies on tooth development are limited, since epithelial stem cells are relatively difficult to obtain and maintain. Human embryonic stem cells (hESCs) and induced pluripotent stem cells (hiPSCs) may be alternative options for epithelial cell sources. To differentiate hESCs/hiPSCs into dental epithelial-like stem cells, this study investigated the hypothesis that direct interactions between pluripotent stem cells, such as hESCs or hiPSCs, and Hertwig’s epithelial root sheath/epithelial rests of Malassez (HERS/ERM) cell line may induce epithelial differentiation. Epithelial-like stem cells derived from hES (EPI-ES) and hiPSC (EPI-iPSC) had morphological and immunophenotypic characteristics of HERS/ERM cells, as well as similar gene expression. To overcome a rare population and insufficient expansion of primary cells, EPI-iPSC was immortalized with the SV40 large T antigen. The immortalized EPI-iPSC cell line had a normal karyotype, and a short tandem repeat (STR) analysis verified that it was derived from hiPSCs. The EPI-iPSC cell line co-cultured with dental pulp stem cells displayed increased amelogenic and odontogenic gene expression, exhibited higher dentin sialoprotein (DSPP) protein expression, and promoted mineralized nodule formation. These results indicated that the direct co-culture of hESCs/hiPSCs with HERS/ERM successfully established dental epithelial-like stem cells. Moreover, this differentiation protocol could help with understanding the functional roles of cell-to-cell communication and tissue engineering of teeth.

## 1. Introduction

Teeth are highly mineralized organs derived from the oral ectoderm and underlying mesenchymal stem cells from the neural crest, which undergo a series of sequential and tightly regulated processes, such as epithelial–mesenchymal interaction [1,2,3]. Recently, the existence of epithelial stem cells has been reported in the human periodontium and deciduous dental pulp that is absolutely required for tooth morphogenesis [4,5]. In adult teeth, these Hertwig’s epithelial root sheath/epithelial rests of Malassez (HERS/ERM) cells derived from the periodontium have unique population and stem cell characteristics [6]. However, there are difficulties in researching the interaction between the epithelium and mesenchyme for dental regeneration due to the very rare population of dental epithelial stem cells and relative difficulty of maintaining the population [7,8]. Therefore, in our previous report, we immortalized primary HERS/ERM cells using the SV40 large T antigen to overcome these limitations [9,10,11].

Pluripotent stem cells, such as embryonic stem cells (ESCs) and induced pluripotent stem cells (iPSCs), have the advantage of being able to differentiate into most cell types because they are relatively non-limiting and non-invasive compared to adult stem cells [12,13]. More importantly, they are easier to use for regenerative therapy because they exhibit a higher proliferation and stem cell potency than oral tissue-derived stem cells cultured in vitro [14]. Despite these advantages, research on the differentiation of dental stem cells has not yet significantly been explored. In the case of mice, the differentiation of iPSCs into epithelial cell-derived ameloblasts has been reported, but no human studies have been conducted [15].

In this study, we used HERS/ERM cell lines as feeder cells for human ESCs (hESCs) and human iPSCs (hiPSCs) to induce their differentiation into dental epithelial-like cells. Additionally, the differentiated iPSC-derived epithelial cells (EPI-iPSC) were immortalized to establish a cell line; subsequently, its dental differentiation potential was confirmed by co-culturing with dental pulp stem cells from human exfoliated deciduous teeth (hdDPSC) capable of amelogenic and odontogenic differentiation [16,17,18,19,20,21,22]. This is the first report of a method for differentiating hESCs and hiPSCs into dental epithelial-like cells via the cellular niche of human dental epithelial cells.

## 2. Results

### 2.1. Differentiation and Characterization of hESCs into Epithelial-Like Stem Cells 

A reciprocal process consisting of 14 days of differentiation on the feeder layer and establishment of the cell line is schematically shown in Figure 1. The HERS/ERM cell line, which was already reported to have similar characteristics to primary HERS/ERM [23], was used as feeder cells for hES/hiPSC. The pluripotent cells were cultured on sheets of the mitomycin C (MMC)-treated HERS/ERM cell line for 14 days in the same medium used for maintaining primary HERS/ERM cells [9]. Sub-culturing was performed in order to select the differentiated epithelial-like cells. 

After 14 days of differentiation, hESCs on HERS/ERM feeder cells grew tightly together in a cuboidal or polygonal shape (Figure 2a). Similar to HERS/ERM, differentiated hESCs expressed epithelial stem cell markers (*E-cadherin, ABCG2, Bmi1, p63,* and *p75*) and embryonic stem cell markers (*Oct-4, Nanog,* and *Sox-2*) (Figure 2b). In addition, immunophenotypes between primary HERS/ERM and differentiated hESCs (EPI-ES) were positive for mesenchymal markers (CD29) and HLA type I, but were negative for hematopoietic cell markers (CD10, CD45, and HLA-DR) and an endothelial cell marker (CD31) (Figure 2c). These data suggested that EPI-ES in this manner has epithelial stem cell characteristics that are comparable to those of primary HERS/ERM.

### 2.2. Establishment of Differentiated Dental Epithelial-Like Stem Cell Lines Derived from hiPSC

In the HERS/ERM feeder system, hiPSCs were differentiated into epithelial-like stem cells that had an epithelial cell morphology, gene expression of epithelial stem cell markers and embryonic stem cell markers, and immunophenotypical characteristics of primary HERS/ERM (Figure 3). Similar to primary HERS/ERM, EPI-iPSCs were considered to be quiescent and have a limited lifespan. Therefore, EPI-iPSCs were transfected with SV40 LT for immortalization. One clone was established via G418 selection. As shown in Figure 4c, three EPI-iPSC cell lines that expressed SV40 were established. Furthermore, all EPI-iPSC cell lines were maintained for more than 15 passages (Figure 4b,c). There were no differences in the morphological and immunophenotypic characteristics of primary HERS/ERM and the EPI-iPSC cell line. The EPI-iPSC cell line expressed epithelial stem cell markers (*E-cadherin*, *ABCG2*, *Bmi1*, *p63*, and *p75*) and embryonic stem cell markers (*Oct-4, Nanog,* and *Sox-2*). It was positive for mesenchymal markers (CD29) and HLA type I, but negative for hematopoietic cell markers (CD10, CD45, and HLA-DR) and an endothelial cell marker (CD31). In addition, the results of chromosomal G-banding indicated a normal karyotype (46 chromosomes, XY) in the EPI-iPSC cell line (Figure 4d–f). To confirm the origin of EPI-iPSC, microsatellite (STR) analysis was performed by the PCR-based microsatellite method. As shown in Figure 4g and Table 1, the EPI-iPSC cell line was derived not from HERS/ERM feeder, but hiPSC. 

### 2.3. Induction of Epithelial-Mesenchymal Transition (EMT) in Dental Epithelial-Like Stem Cell Lines Derived from hiPSCs

In previous reports, primary HERS/ERM and its cell line could acquire a mesenchymal phenotype through EMT via TGF-β1 [9,24,25]. The EPI-iPSC cell line was similarly induced by TGF-β1 to investigate whether it would also acquire mesenchymal phenotypes through EMT. As a result, the EPI-iPSC cell line exhibited mesenchyme-like morphological changes, such as elongated shapes and a loss of cell-to-cell contact (Figure 5a). To further elucidate the effect of EMT, the gene expression of *E-cadherin*, *N-cadherin*, and *Vimentin* was examined. After EMT induction, the EPI-iPSC cell line demonstrated a down-regulated expression of E-cadherin. On the other hand, expressions of N-cadherin and Vimentin were significantly up-regulated. (Figure 5b). These data suggested that the EPI-iPSC cell line could acquire mesenchymal phenotypes through EMT.

### 2.4. Differentiation Potential of Differentiated Dental Epithelial-Like Stem Cell Lines Derived From hiPSC

To observe the synergetic effect of EPI-iPSC and hdDPSC, co-culture was performed with or without osteogenic medium for 20 days. The expression of ameloblast/odontoblast markers was measured with qRT-PCR and a western blot. Amelogenin, the major structural protein of the enamel organic matrix, was notably increased in EPI-iPSC alone or the co-culture group when odontogenic differentiation was induced. Ameloblast matrix protein expression, including enamelin and the proteinase KLK4, was only upregulated in the co-culture group with the induction medium compared with hdDPSC or EPI-iPSC alone. An early and proliferative stage marker of mineralization, such as Runx2 and BSP of the matrix maturation marker, was increased in the co-culture group with a5OB. OCN, which is an even later stage marker than matrix maturation, was expressed in EPI-iPSC alone and the co-culture group. DMP1, essential in the maturation of an odontoblast, was expressed significantly higher in both hdDPSC alone and the co-culture group than in EPI-iPSC alone. DSPP is a key marker for the odontoblast lineage in this study because DSPP is necessary for early differentiation and late dentin mineralization. The mRNA expression of DSPP was only highly expressed by co-culturing it with the odontogenic induction medium (Figure 6a). 

Additionally, these expression patterns were further confirmed by a western blot. Consistent with gene expression, dentin sialoprotein (DSPP) proteins were highly expressed when co-culturing with hdDPSCs and EPI-iPSC compared with hdDPSC or EPI-iPSC alone (Figure 6b). Although DSPP was detected at day 12 in hdDPSC or EPI-iPSC alone under odontogenic induction, the expression was lower than in the co-culture group. The mineralized nodule formation was evaluated with Alizarin red S staining on day 12, 16, and 20 [26]. EPI-iPSC and hdDPSC were used as controls. The co-culture group in osteogenic medium exhibited highly increased nodule formation compared to controls at day 12. The size of mineralized nodules was markedly increased on day 20, while EPI-iPSC or hdDPSC alone showed fewer mineralized nodules (Figure 6c).

## 3. Discussion

The epithelial–mesenchymal interaction plays an important role in the development of various organs, including mammalian teeth [27,28]. Many in vitro and in vivo studies in this area show that tooth formation is regulated by epithelial–mesenchymal interaction, during which ameloblasts are involved in the formation of enamel, while mesenchyme-derived odontoblasts form dentin [29]. Therefore, strategies involving epithelial–mesenchymal interaction have successfully regenerated tooth-like organoid structures [8,20,30]. However, clinical applications using such bioengineering techniques and further tooth regeneration studies are limited, primarily due to the lack of appropriate sources of dental epithelial stem cells. 

Adult dental stem cells have been successfully applied in tissue engineering research, including possibilities to generate dentin pulp complexes or whole teeth out of isolated cells [31,32]. However, it has been difficult to study odontogenic or amelogenic differentiation because HERS/ERM cells disappear during tooth eruption due to apoptosis. For this reason, most studies on dental development have been performed with mouse and rat incisors. Therefore, an alternative human source of primary dental epithelial cells is absolutely required, in order to establish a practical and reliable method for regenerative dentistry. Notably, accessibility to the source of HERS/ERM cells, which reside as a rare population in the periodontium and proliferate very slowly, is challenging. We previously reported how to overcome the limited life-span of primary dental epithelial stem cells through SV40 transfection, which inactivates the tumor suppressor p53 and Rb pathways. The immortalized HERS-SV40 and primary HERS/ERM showed the same morphological characteristics, immunophenotypes, and gene expression profiles [9]. However, these cell lines are established from primary HERS/ERM cells, which are generally not easy to obtain from teeth extracted from adults [33].

Over the past decade, numerous studies have indicated that the intrinsic properties of ESCs and iPSCs offer great opportunities for cell therapy, and the cells have been considered to be promising sources for dental regenerative medicine. Recent reports have described the in vitro differentiation of ESCs or iPSCs into dental epithelial cells. Ning et al., developed differentiation strategies that induced mouse ESCs into dental epithelial cells by culturing the cells in ameloblast serum-free conditioned medium [34]. The dental epithelial-like cells expressed ameloblast-specific markers, such as *cytokeratin 14, amelogenin,* and *ameloblastin*. Arakaki et al., demonstrated that mouse iPSCs (miPSCs) had the capacity to differentiate into dental tissues like ameloblasts by co-culturing miPSC and dental epithelium. This resulted in epithelial-like cells that expressed epithelial cell markers such as *p63* and *cytokeratin 14*, as well as ameloblast markers such as *ameloblastin* and *enamelin* [15,35]. In a recent study, the addition of *neurotrophin-4* during the formation of the embryonic body significantly triggered the upregulation of epithelial markers such as *p63* and *CK14* in miPSCs [23]. Others, like Kim et al., have succeeded in differentiating miPSCs into dental epithelial cells using conditional media of SF2 dental epithelial cells by adjusting the embryonic body [36]. It has also been reported that hiPSCs derived from cells other than dental tissues could generate epithelial sheets and enamel-secreting ameloblasts [37]. However, due to the properties of the epithelial sheet, continuous and long-term culture is limited, which is not ideal for a cell source. Despite having various reports on novel methods for generating dental epithelial-like cells from ES or iPSC, no report has yet described the differentiation of human ESCs or human dental stem cell-derived iPSCs.

Dental pulp stem cells (DPSCs) are a part of the dental mesenchymal stem cell source, derived from the cranial neural crest, and differentiate into dentin forming odontoblast. Moreover, it is well-known that reciprocal communication between dental epithelial and mesenchymal cells leads to tooth morphogenesis [20]. Epithelial stem cells such as HERS/ERM cells then subsequently differentiate into enamel forming ameloblasts, while mesenchymal stem cells form dentin-forming odontoblasts and dental pulp cells [23].

In recent years, many research groups have successfully created co-culture systems based on the speculation that intricate signaling crosstalk may be established among heterogeneous cell populations and possibly regulated by direct cellular interaction or paracrine signaling [38]. When miPSCs were co-cultured with dental epithelium, miPSCs differentiated into ameloblasts via paracrine signaling of BMP2 and BMP4 secreted by dental epithelial cells [15]. Likewise, this study showed that hESCs and hiPSCs derived from hdDPSCs have the potential to generate dental epithelial-like stem cells using our co-culture system. This strategy may have a direct application in tooth regeneration. The co-culture system was established as follows.

First, hESCs and hiPSCs were co-cultured with HERS-SV40 to differentiate into dental epithelial-like stem cells. The co-culture of hiPSC with epithelial stem cells resulted in the formation of cells with an epithelial-like morphology that expressed stemness and epithelial stem cell markers. RT-PCR showed that they expressed *ABCG2, EpCAM, P63, P75*, and *E-cadherin. ABCG2*, a member of the ABC transporter family, has been proposed as a universal stem cell marker [39,40]. *BMI1* has an essential role in embryogenesis, but its roles in epithelial stem cells have not been fully specified [41,42,43]. *P75* is highly expressed in the basal cells of many human epithelial tissues, and *P63* is associated with proliferative potential in human keratinocytes [44,45,46]. Several studies have shown that *E-cadherin* has two functions in teeth; one is to maintain epithelial stem cells in the cervical loop niches, and the other is to enable proper cell migration [47]. As shown in Figure 2 and Figure 3, both EPI-ES and EPI-iPSC expressed epithelial stem cell markers *(ABCG2, EpCAM, P63, P75,* and *E-cadherin)* similar to primary HERS/ERM cells. These data suggested that EPI-ES and EPI-iPSC acquired epithelial stem cell characteristics that were comparable to those of primary HERS/ERM cells.

Next, we established an immortalized epithelial-like stem cell line from EPI-iPSCs to produce sufficient cell numbers through long-term culture for research purposes. The morphological and immunophenotypic characteristics were maintained, and the cells showed the same surface protein expression pattern as HERS/ERM cells. More specifically, CD29 was expressed in most of the mesenchymal stem cell markers, while CD10 was not. Moreover, hematopoietic markers such as CD45 and HLA-DR were not expressed, while the human major histocompatibility complex (MHC) marker HLA-I was expressed in most cells. The expression of epithelial stem cell and embryonic stem cell markers was maintained in the immortalized HERS/ERM cells. Additionally, immortalized EPI-iPSC acquired mesenchymal phenotypes through EMT by TGF-β1 (Figure 5). These results indicate that the EPI-iPSC cell line had similar characteristics to primary HERS/ERM cells and might be a prospective cell source for understanding the biological functions of dental development.

Despite these results, it is considered that TGF-β1 may not be a sufficient inducer of EMT in HERS/ERM cells. According to other groups, unexpectedly, TGF-β1-treated HERS/ERM cells formed a tiny amount of cementum-like tissue through EMT [48]. Furthermore, our previous study showed that the expression of E-cadherin was not decreased in either the primary HERS/ERM cells or immortalized HERS/ERM cells [9]. Based on this, additional experiments are needed to obtain epithelial stem cells with stable EMT characteristics. 

It is well-known that protein arginine methyltransferase 1 (PRMT1) enables TGF-signaling to regulate EMT and the stemness of epithelial cells through SMAD6 methylation at the bone morphogenic protein (BMP) receptor complex, and thus promotes the TGF-induced EMT and epithelial stem-cell generation [49]. Moreover, by using media containing bFGF, VEGF, EGF, and R3-IGF-1, hESCs could differentiate into an epithelial sheet and obtain EMT characteristics [50]. 

Additionally, we evaluated the effects of EPI-iPSC on odontogenic differentiation and the bone formation of hdDPSC in vitro for 20 days. Epithelial stem cells are well-known to give rise to enamel-forming ameloblasts, while mesenchymal stem cells generate dentin-forming odontoblasts and dental pulp cells. As expected, typical ameloblast-related gene markers (*Amelogenin, Enamelin,* and *KLK-4*) were significantly increased in the co-culture group under odontogenic induction. Furthermore, both early mineralization genes (*Runx2* and *BSP*) and late mineralization genes (*OCN, DMP1,* and *DSPP*) were highly expressed in the co-culture group [51]. Therefore, it has been demonstrated that the in vitro co-culture of EPI-iPSC/hdDPSC showed more active odontogenic differentiation, indicated by the increased protein levels of DSPP. Coincident with our data, it is reported that the DSPP was expressed when co-culturing with DPSCs and dental epithelial cells compared with DPSC or epithelial cells alone. Furthermore, by detecting epithelial markers, it was clear that the DSPP expressing odontoblasts that formed when co-cultured among two cell types were not epithelium, but DPSC [15]. In addition, when mineralized calcium nodules were stained by Alizarin Red S for analyzing the mineralization efficiency, the intensity was highly accelerated in the co-culture group with odontogenic induction medium compared to hdDPSC or EPI-iPSC alone. Taken together, these results demonstrate that interactions between hdDPSC and EPI-iPSC efficiently induced ameloblast or odontoblast lineages in a synergistic manner. 

The interaction of HERS/ERM cells and hESCs or iPSCs sufficiently provides an environment to induce differentiation into cells that exhibit several features, such as natural HERS/ERM in terms of the expression pattern of marker genes, and potential for further differentiation towards ameloblasts. Therefore, it is important to understand the communication between cells through cytokine gradients, such as paracrine and juxtacrine effects, in order to investigate stem cell differentiation niches [52]. Furthermore, advanced approaches to differentiation conditions are needed to improve the efficiency of protocols and to eliminate the possibility of cell contamination by a direct-culture system.

Taken together, by co-culturing HERS/ERM cells and human pluripotent stem cells, we have identified human ESCs and iPSCs as practical sources of dental epithelial stem cells. Our results suggest a novel therapy for regenerative medicine and teeth regeneration involving the established EPI-iPSC. For absolute autologous dental recovery, dental mesenchymal stem cell (MSC) derived from iPSCs are also required in further studies.

## 4. Materials and Methods

### 4.1. Cell Preparation and Culture Conditions

The protocol was approved by the Institutional Review Board of School of Dentistry, Seoul National University (S-D20080009, 29 August 2017). Written informed consent was obtained from all participants. H9 hESCs were purchased from a commercial source (WiCell Research Institute, Madison, MA, USA). Human iPSCs were established from hdDPSCs using an episomal vector system. Both cell types were cultured on 0.2% gelatin-coated (Bio Basic Inc, Markham, Ontario, Canada) four-well plates on mouse embryonic fibroblast (MEF) feeder layers, and were mechanically transferred every four to five days. Undifferentiated hESCs/hiPSCs were chosen for each further passaging and cut away from the feeders as small clumps. The hESCs/hiPSCs were cultured in DMEM/F12, which was supplemented with 20% (*v*/*v*) knockout serum replacement (KSR), 10 mM nonessential amino acids (all from Gibco-Invitrogen, Cergy-Pontoise, France), 50 mM β-mercaptoethanol (Sigma Aldrich, St. Louis, MO, USA), and 4 ng/mL basic FGF (Gibco-Invitrogen, Carlsbad, CA, USA). Cultures were maintained at 37 °C in a humidified atmosphere containing 5% CO_2_.

### 4.2. Differentiation of Human ESCs and iPSCs into Epithelial-Like Cells

The HERS/ERM cell line was used as feeder cells to induce the differentiation of hESCs/hiPSCs into epithelial stem cells. Mitomycin C (MMC, Sigma Aldrich, St. Louis, MO, USA) was treated at a concentration of 10 μg/mL for 1 h 30 min to inactivate HERS/ERM mitotically. MMC-treated HERS/ERM cells were trypsinized, and 10^6^ cells were cultured in 0.1% gelatin-coated 35 mm cell culture dishes. After 24 h, hESCs/hiPSCs clumps were transferred onto the feeder layers. Two weeks later, differentiated epithelial-like cells were dissociated with accutase (Merck Millipore, Billerica, MA, USA), and the cells were cultured on a new dish. The culture medium used was serum-free keratinocyte growth medium-2 (KGM-2; Lonza, Rock Island, ME, USA) with supplements provided (Lonza, Rock Island, ME, USA). The medium was changed every two days. Cells were sub-cultured at 70% confluence.

### 4.3. SV40 LT Transformation-Mediated Immortalization

Epithelial-like cells were used for transfection. The cells (15,000 cells/cm^2^) were transfected with 2 μg of a pcDNA 3.1 (+) plasmid containing the SV40 large T antigen (SV40) using the X-tremeGENE 9 DNA transfection reagent (Roche, Indianapolis, IN, USA), according to the manufacturer’s instructions. The transfection mixture was cultured for two days in KGM-2, and G418 (Cellgro Mediatech, Washington, DC, USA) was then added to the medium at a concentration of 100 μg/mL. The supplemented G418 was changed every two days. Two weeks after G418 selection, independent colonies were observed and maintained with G418-free KGM-2 culture medium. After another two weeks, cells were transferred to new culture dishes. 

### 4.4. Immunofluorescent Staining

To detect the overexpression of SV40, epithelial-like cell lines were fixed with 4% paraformaldehyde in Dulbecco’s phosphate-buffered saline (DPBS, Welgene, Daegu, Korea) for 10 min. Cells were washed with DPBS and then permeabilized with 0.5% Triton X-100 (Bio-rad, Hercules, CA, USA). Non-specific binding events were minimized by 10% normal goat serum (Jackson Immuno Research Laboratories, West Grove, PA, USA) for 1 h at room temperature (RT). Immunostaining was performed with the antibody against the SV40 LT rabbit polyclonal antibody (Santa Cruz Biotechnology, Santa Cruz, CA, USA) at a dilution of 1:50 for 1 h. Alexa Fluor 488 Goat anti-rabbit IgG was used as a secondary antibody (Invitrogen, Carlsbad, CA, USA) at a dilution of 1:700. Nuclear counterstaining was performed with DAPI (Sigma–Aldrich, St Louis, MO, USA) at a dilution of 1:1000. Images of stained cells were observed using an inverted fluorescence microscope (TE2000-U, Nikon Eclipse, Tokyo, Japan).

### 4.5. Microsatellite (STR) Analysis

Genomic DNA was extracted from the iPSCs, epithelial-like cells, and HERS/ERM cells using the genomic DNA Purification Kit (LaboPassTM Tissue mini, Cosmogenetech, Seoul, Korea). Microsatellite (STR) analysis was performed by Macrogen Inc. (Seoul, Korea).

### 4.6. Total RNA Preparation and Reverse Transcription

HERS/ERM and epithelial-like cells were transferred to RNAlater (Invitrogen, Carlsbad, CA, USA) and stored at −20 °C. Cells were washed with DPBS supplemented with 2% FBS. Total RNA was subsequently extracted using the RNeasy Mini Kit (Qiagen, Valencia, CA, USA). DNase I treatment was performed for the removal of genomic DNA contamination using an RNase-free DNase set (Qiagen, Valencia, CA, USA). The total RNA (2 μg) was reverse-transcribed with amfiRivert cDNA synthesis Platinum Master Mix (GenDEPOT, Barker, TX, USA). The reverse transcription was carried out at 25 °C for 5 min and then 42 °C for 60 min. Finally, heat inactivation of the enzyme at 70 °C for 15 min was performed. The RT products were diluted 10-fold and stored at −20 °C until PCR amplification.

### 4.7. Semi-Quantitative PCR

PCR amplification was conducted with i-MAXII (Intron, Seongnam, Gyeonggi, Korea), and standard PCR conditions were as follows: 2 min at 94 °C, followed by cycles of 20-s denaturation at 94 °C, 15-s annealing at each temperature, and 30-s extension at 72 °C. A final extension step was performed at 72 °C for 5 min. Primer sequences and each annealing temperature are shown in the Table 2. For the analysis of PCR products, DNA gel electrophoresis was carried out in 1.5% agarose gel with 0.01 mg/mL ethidium bromide and quantified by Bio-profil X press zoom 2000 (Vilber Lourmat, Marne la Vallée, France).

### 4.8. Flow Cytometry

For the characterization of epithelial-like cells compared with HERS/ERM, fluorescence-activated cell sorting (FACS) was performed. The cells were detached and washed with DPBS supplemented with 2% FBS and then fixed with 4% paraformaldehyde at RT for 10 min. After washing with DPBS, 10,000 cells were incubated with fluorescently conjugated antibodies for 20 min at 4 °C. The following antibodies were used: FITC-conjugated mouse anti-human CD31, HLA-DR, HLA-I, PE-conjugated mouse anti-human CD10, CD29, and APC-conjugated mouse anti-human CD45 (all from BD Pharmingen, San Diego, CA, USA). The fluorescence intensity was measured by a FACS Calibur, and data were analyzed with FlowJo software (Tree Star, San Carlos, CA, USA).

### 4.9. Epithelial-Mesenchymal Transition (EMT) Induced by Transforming Growth Factor-β1 (TGF-β1)

To investigate the EMT of epithelial-like cells and HERS/ERM via TGF-β1, the cells were treated with 20 ng/mL TGF-β1 (Peprotech, Rocky Hill, NJ, USA) for 48 h.

### 4.10. Real-Time PCR

The cDNA obtained from epithelial-like cells and HERS/ERM was amplified in a reaction mixture (20 μL) containing 10 μL of THUNDERBIRD SYBR qPCR Mix (Toyobo, Suita, Osaka, Japan) and each primer at 0.5 mM. The conditions for the PCR and primer sequences are listed in Table 2. Real-time PCR was performed with the CFX Connect Real-Time PCR Detection System (Bio-Rad, Hercules, CA, USA). The copy numbers of mRNA were standardized to those of glyceraldehyde-3-phosphate dehydrogenase (GAPDH). GAPDH was used as internal normalization controls for mRNA.

### 4.11. Odontogenic Differentiation

The odontogenic differentiation capacity was determined by analyzing the selective differentiation marker expression. The conditions used for odontogenic differentiation of the EPI-iPSC cell line with hdDPSCs were as shown in Table 3. To evaluate calcium nodule formation, the differentiated cells were stained with Alizarin red S.

### 4.12. Western Blotting

The differentiated cells were lysed with RIPA buffer to extract protein. A total of 10 μg of each protein extract was electrophoresed on a sodium dodecyl sulfate-polyacrylamide (SDS-PAGE) gel and then transferred to a PVDF membrane. Blocked membranes were incubated with primary anti-DSPP (Santa Cruz Biotechnology, Santa Cruz, CA, USA) antibodies, followed by horseradish peroxidase-conjugated secondary antibodies. The chemiluminescent intensity of immunoblotted bands was visualized using a ChemiDoc Imaging System (Bio-Rad, Hercules, CA, USA). The intensity of each band was normalized to the β-actin band intensity (Novus Biologicals, Littleton, CO, USA).

### 4.13. Statistical Analysis

All data are presented as means ± standard deviations (SDs) of the values obtained in experiments performed at least in triplicate. Statistical analysis was performed by one-way ANOVA followed by a least-significant difference (LSD) post-hoc test with Prism 8 software (GraphPad, San Diego, CA, USA). Statistically significant differences between data were reported if *p* < 0.05.

## Figures and Tables

**Figure 1 ijms-21-04384-f001:**
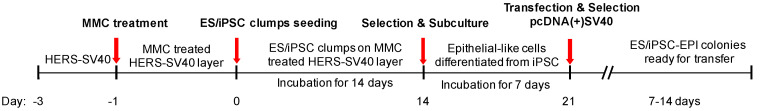
Experimental scheme for the differentiation of human pluripotent stem cells into dental epithelial-like cells. Mitomycin C (MMC)-treated Hertwig’s epithelial root sheath/epithelial rests of Malassez (HERS/ERM) cells were used as a feeder layer for epithelial-like cell differentiation. Two weeks after the seeding of human induced pluripotent stem cell (hiPSC) clumps on the feeder layer, differentiated epithelial-like cells were collected and characterized. To overcome the limited lifespan of epithelial-like cells, the cells were transfected with an SV40 gene. Immortalized cell lines selected through G418 were characterized by morphology and genetic/immunophenotypic characteristics.

**Figure 2 ijms-21-04384-f002:**
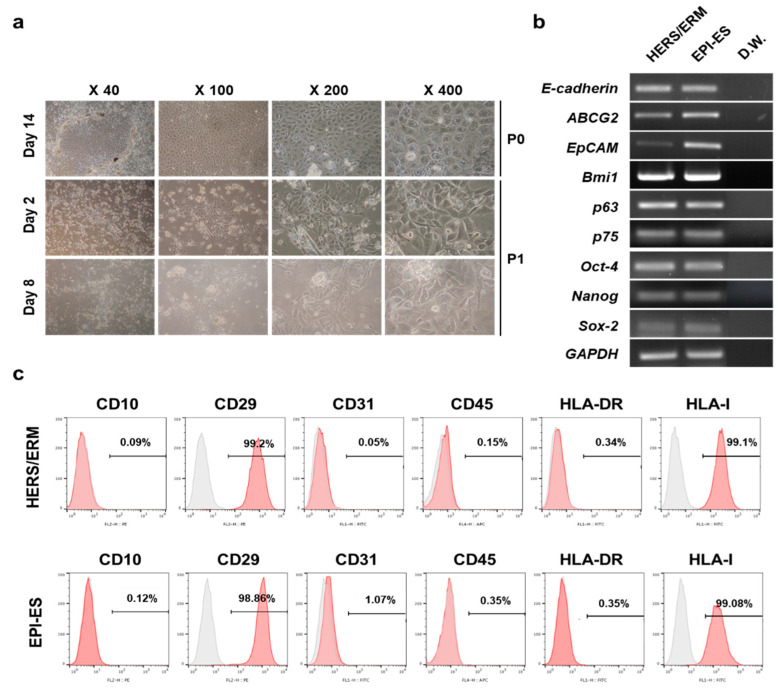
Differentiation of human embryonic stem cells (hESC) into dental epithelial-like stem cells (EPI-ES). (**a**) Morphological change during hESC differentiation into EPI-ES. After 14 days on the feeder layer, typical epithelial cell-like cuboidal- or polygonal-shaped appearances were observed around the hESC clumps. During two to eight days after subculture, cells with these morphologies underwent colony-forming proliferation. (**b**) Gene expression of EPI-ES after 14 days of induction. All samples, as well as HERS/ERM, were positive for epithelial stem cell markers and stemness markers. (**c**) Expression of surface antigens of EPI-ES. Both primary HERS/ERM and all epithelial-like cell lines were positive for mesenchymal markers (CD29) and HLA type I, but negative for hematopoietic cell markers (CD10, CD45, and HLA-DR) and an endothelial cell marker (CD31). All data were replicated three times.

**Figure 3 ijms-21-04384-f003:**
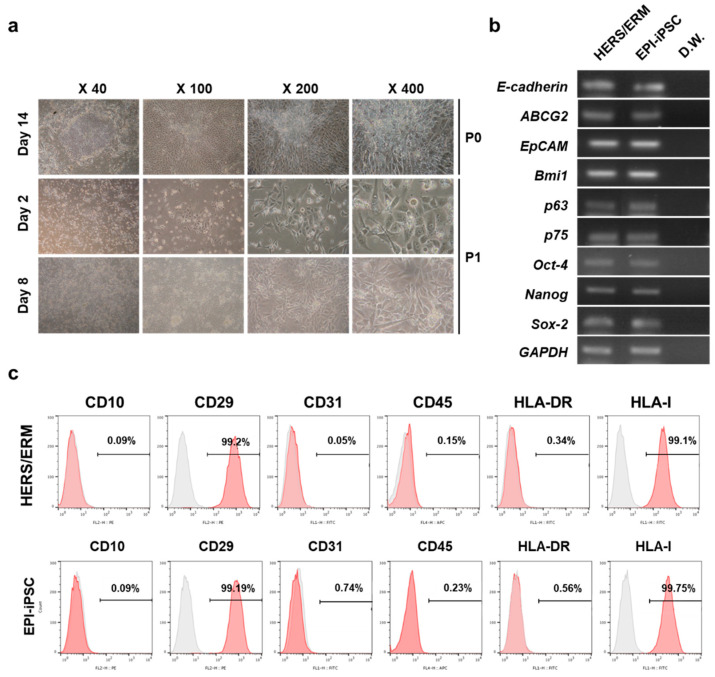
Differentiation of human iPSC (hiPSC) into dental epithelial-like stem cells (EPI-iPSC). (**a**) Morphological change of hiPSC after differentiation. On day 14, hiPSC had an epithelial cell-like appearance and colony-forming proliferation during subculture. (**b**) Expression of genes related to the epithelial stem cells in EPI-iPSC after 14 days of induction. EPI-iPSC was positive for E-cadherin, *ABCG2*, *EpCAM*, *Bmi1*, *p63*, and *P75*, which are epithelial stem cell markers, and *Oct-4*, *Nanog*, and *Sox-2,* which are stemness-related markers. (**c**) Fluorescence-activated cell sorting (FACS) analysis of EPI-iPSC. EPI-iPSC was positive for mesenchymal markers (CD29) and HLA type I, but negative for hematopoietic cell markers (CD10, CD45, and HLA-DR) and an endothelial cell marker (CD31). All data were replicated three times.

**Figure 4 ijms-21-04384-f004:**
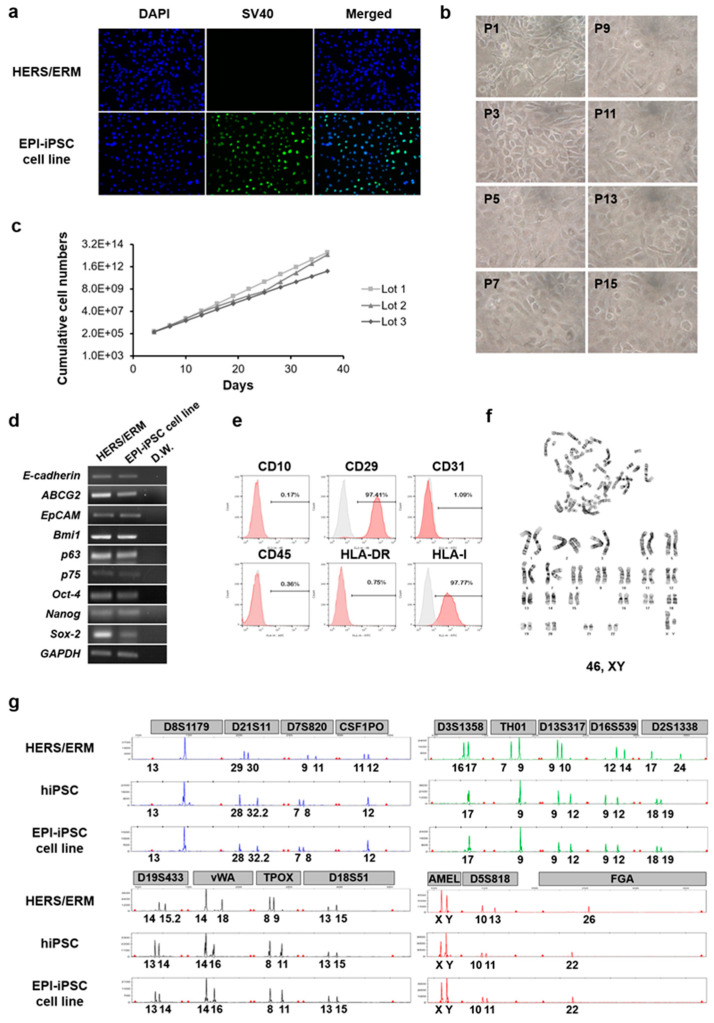
Characterization of dental epithelial-like stem cell lines derived from hiPSC. (**a**) Immunofluorescence staining for the expression of SV40 in the EPI-iPSC cell line. Primary HERS/ERM cells did not express SV40, whereas the established EPI-iPSC cell line expressed SV40. (**b**) Morphology and passaging of the EPI-iPSC cell line. EPI-iPSC-SV40 showed the typical epithelial cell-like shape and clonal expansion until passage 15. The morphology was maintained through subculture. Magnifications are at 400×. (**c**) Growth of three EPI-iPSC-SV40 lines. Cumulative cell numbers of EPI-iPSC showed that they maintained stable proliferation for 40 days. (**d**) Expression of epithelial stem cell and stemness-related genes in the EPI-iPSC cell line (passage 10). EPI-iPSC cell line was positive for *E-cadherin*, *ABCG2*, *EpCAM*, *Bmi1*, *p63*, and *P75*, which are epithelial stem cell markers, and *Oct-4*, *Nanog*, and *Sox-2,* which are stemness-related markers. (**e**) FACS analysis of the EPI-iPSC cell line (passage 10). EPI-iPSC was positive for CD29 and HLA-I, and negative for CD10, CD45, HLA-DR, and CD31. (**f**) Karyotype of the EPI-iPSC cell line. The EPI-iPSC cell line at passage 10 showed a normal karyotype with 46, XY. (**g**) Origin of the EPI-iPSC cell line. Microsatellite (STR) analysis, which is a PCR-based microsatellite method, showed that the differentiated EPI-iPSC cell line was derived from hiPSC. All data were obtained from three replicates.

**Figure 5 ijms-21-04384-f005:**
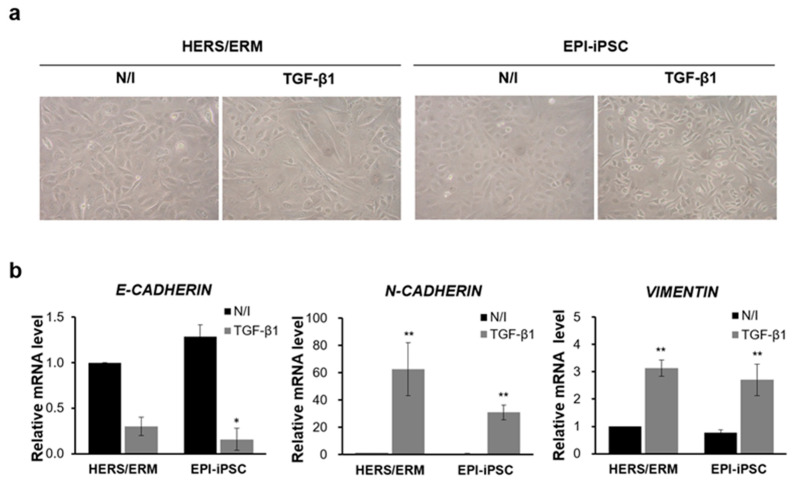
Epithelial-mesenchymal transition (EMT) of HERS/ERM cells and the EPI-iPSC cell line. The EMT was induced by TGF-β1 for 48 h. (**a**) Morphology of the EPI-iPSC cell line after 48 h of TGF-β1 treatment. All of these cells lost epithelial cell polarity and cell-to-cell contact. (**b**) EMT-related gene expression of the EPI-iPSC cell line after EMT induction. When all cell types were treated with TGF-β1, the gene expression of N-cadherin and Vimentin was increased in primary HERS/ERM and epithelial-like cells. However, the levels of E-cadherin were decreased. All data shown are the mean ± S.D. from the levels of three replicates. Data are presented as the mean ± SD, *n* = 6 per group. ** *p* < 0.01, * *p* < 0.05. N/I: no induction.

**Figure 6 ijms-21-04384-f006:**
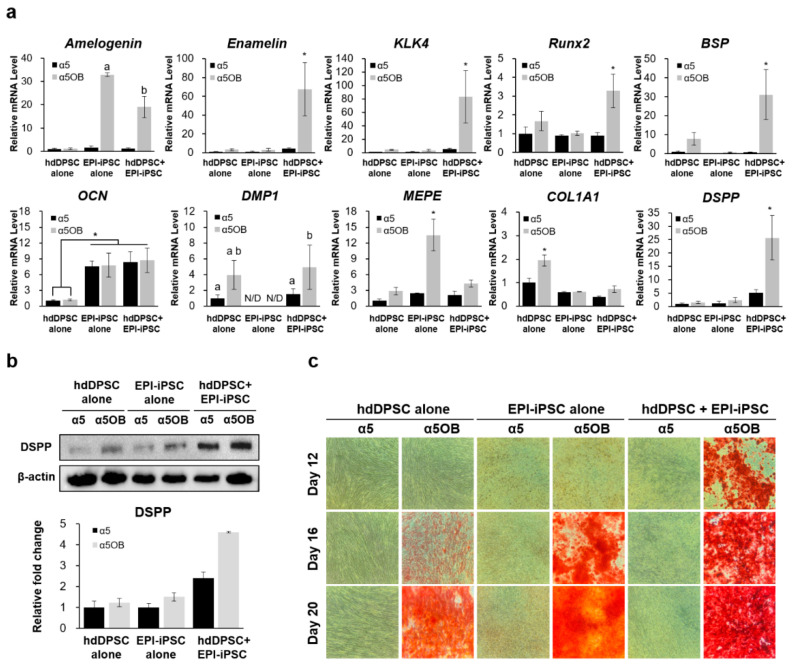
Effect of the EPI-iPSC cell line on the odontogenic differentiation and mineralization of hdDPSC. EPI-iPSC and hdDPSC were co-cultured with or without a differentiation medium. The hdDPSC and EPI-iPSC alone group was analyzed as a control group. (**a**) Expression of ameloblast markers (*Amelogenin*, *Enamelin*, and *KLK4*) and odontoblast markers (*Runx2*, *BSP, OCN, DMP1*, *MEPE*, *Col1A1*, and *DSPP*) was analyzed by real-time PCR after eight days in culture. (**b**) Expression patterns of dentin sialoprotein (DSPP) were evaluated by a western blot after 12 days in culture. (**c**) Mineralized nodule formation of co-culture with EPI-iPSC and hdDPSC in vitro was evaluated by Alizarin red S staining on days 12, 16, and 20. Data are presented as the mean ± SD, *n* > 3 per group. ^a,b,^* *p* < 0.05. N/D: Not detected.

**Table 1 ijms-21-04384-t001:** STR analysis showed that the EPI-iPSC cell line matched human iPSCs.

HERS-SV40		Human iPSCs		EPI-iPSC	
STR Locus	Genotype	STR Locus	Genotype	STR Locus	Genotype
D8S1179	13	D8S1179	13	D8S1179	13
D21S11	29, 30	D21S11	28, 32.2	D21S11	28, 32.2
D7S820	9, 11	D7S820	7, 8	D7S820	7, 8
CSF1PO	11, 12	CSF1PO	12	CSF1PO	12
D3S1358	16, 17	D3S1358	17	D3S1358	17
TH01	7, 9	TH01	9	TH01	9
D13S317	9, 10	D13S317	9, 12	D13S317	9, 12
D16S539	12, 14	D16S539	9, 12	D16S539	9, 12
D2S1338	17,24	D2S1338	18, 19	D2S1338	18, 19
D19S433	14, 15.2	D19S433	13, 14	D19S433	13, 14
vWA	14, 18	vWA	14, 16	vWA	14, 16
TPOX	8, 9	TPOX	8, 11	TPOX	8, 11
D18S51	13, 15	D18S51	13, 15	D18S51	13, 15
AMEL	X, Y	AMEL	X, Y	AMEL	X, Y
D5S818	10, 13	D5S818	10, 11	D5S818	10, 11
FGA	26	FGA	22	FGA	22

**Table 2 ijms-21-04384-t002:** Sequences of primers used for indicated target genes.

Target Gene	5′ Oligonucleotide	3′ Oligonucleotide
*GAPDH*	GAT GCT GGC GCT GAG TAC G	GCT AAG CAG TTG GTG GTG C
*N-cadherin*	ACA GTG GCC ACC TAC AAA GG	CCG AGA TGG GGT TGA TAA TG
*E-cadherin*	TGC CCA GAA AAT GAA AAA GG	GTG TAT GTG GCA ATG CGT TC
*Vimentin*	TCT ACG AGG AGG AGA TGC GG	GGT CAA GAC GTG CCA GAG AC
*ABCG2*	CCA CAG GTG GAG GCA AAT CT	TCG CGG TGC TCC ATT TAT CA
*EpCAM*	GCT GGC CGT AAA CTG CTT TG	ACA TTT GGC AGC CAG CTT TG
*BMI1*	CAG CCC AGC AGG AGG TAT TC	GGA TGA GGA GAC TGC ACT GG
*P63*	ATG TTG TAC CTG GAA AAC AAT GC	GTG ATG GAG AGA GAG CAT CGA A
*P75*	ACC GAG CTG GAA GTC GAG	CTC ACC GCT GTG TGT GTA C
*SOX2*	GAC TTC ACA TGT CCC AGC AC	GGG TTT TCT CCA TGC TGT TT
*OCT4*	ACC CCT GGT GCC GTG AA	GGC TGA ATA CCT TCC CAA ATA
*NANOG*	CCT ATG CCT GTG ATT TGT GG	TTC TCT GCA GAA GTG GGT TG
*Amelogenin*	AAG AAC CAT CAA GAA ATG GGG	TGA TAT AAC CAG GGT GCC CA
*Enamelin*	TCC ACG GAA ATC CTC AGC AC	GGG GGT TGA GCT TCC TCT TC
*KLK4*	GAT CGC TCG TCT CTG GTA GC	GAG TTC TGG AAA CAG TGT GCG
*Runx2*	CCC AGT ATG AGA GTA GGT GTC C	GGG TAA GAC TGG TCA TAG GAC C
*BSP*	AAG GCT ACG ATG GCT ATG ATG GT	AAT GGT AGC CGG ATG CAA AG
*OCN*	ATC CTT TGG GGT TTG GCC TAC	GCC AAT AGG GCG AGG AGT G
*DMP1*	ACA GGC AAA TGA AGA CCC	TTC ACT GGC TTG TAT GG
*MEPE*	CAA GAA GCC AGG TAT TCT GAA GG	TGT GGT TGA AAT GTT GGT GCT
*Col1a1*	CAA AAA ATG GGA GAC AAT TTC ACA	TCA TGT TCG GTT GGT CAA AGA T
*DSPP*	GCC AGA GCA AGT CTG GTA ACG GT	TGT CTC TGC AGG AGT TAG GTC TTG GT

**Table 3 ijms-21-04384-t003:** Odontogenic differentiation medium.

Components	Contents of α5 (Control Group)	Contents of α5OB (Differentiation Group)
Alpha MEM	-	-
FBS	5%	5%
Antibiotics-antimycotics	1%	1%
Dexamethasone	-	0.1 µM
B-glycerophosphate	-	10 mM
L- ascorbic acid phosphate	-	50 µg/mL

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
