# Peer review of "Differentiation and Establishment of Dental Epithelial-Like Stem Cells Derived from Human ESCs and iPSCs"

_ijms, 2020, doi:10.3390/ijms21124384_

Round 1
Reviewer 1 Report
The authors have responded to the critiques and examined DSPP expression in hdDPSC+EPI-iPSC group with hdDPSC only and EPI-iPSC only groups as controls.
Reviewer 2 Report
I thank the authors for their answer to my last review.
This manuscript is a resubmission of an earlier submission. The following is a list of the peer review reports and author responses from that submission.
Round 1
Reviewer 1 Report
The authors used HERS/ERM cell line to induce the differentiation and establishment of dental epithelial-like stem cells derived from human ESCs and iPSCs. These epithelial-like cells showed similar morphological and immunophenotypic characteristics and similar gene markers as those from HERS/ERM cells. The authors further developed an immortalized EPI-iPSC cell line. When EPI-iPSC cell line was co-cultured with dental pulp stem cells, the authors observed increased amelogenic and odontogenic gene expression.
In general, this is an interesting manuscript. The authors overcame the issue of a rare population of the dental epithelial stem cells in the field. They were able to generate epithelial-like stem cells and established an immortalized EPI-iPSC cells line. Furthermore, they could use this cell line to co-culture dental pulp stem cells to investigate epithelial and mesenchymal interactions . However, there is a major concern about Fig. 6. For examining the effect of EPI-iPSC cell line on odontogenic differentiation and mineralization of hdDPSC, the authors should include EPI-iPSC control and hdDPSC control with or without a differentiation medium.
In addition, there are several critiques for this manuscript listed below.
Line 19: Co-cultured and differentiated epithelial-like stem cells derived from hES (EPI-ES) and hiPSC (EPI-iPSC) had morphological and immunophenotypic characteristics of HERS/ERM cells, as well as similar gene expression.
Comment: Please improve this statement.
Line 24: STR
Comment: Please provide full name for STR. For example, short tandem repeat (STR).
Line 67: MMC
Comment: please provide full name for MMC, such as mitomycin C (MMC)
Line 162: The expression of ameloblast markers such as Amelogenin, Enamelin, and KLK-4 was detected when odontogenic differentiation was induced (Figure 6a).
Comment: Is any difference between with and without differentiation medium odontogenic differentiation in inducing the expression of ameloblast markers? It seems to this reviewer that the differentiation medium induced more expression of ameloblast markers. If so, the authors need to clarify this point.
Line 163: Runx2, ONN, OCN, and Col1A1 were expressed in both cell cultures with and without differentiation induction medium. BSP, DMP1, and MEPE expression were increased in the differentiation condition (α5OB) compared to the control group (α5).
Comment: Actually from Fig. 6a, DSPP expression was increased in the differentiation condition (α5OB) compared to the control group (α5).
Line 170: The expression of DSPP protein increased on day 20, especially in the co-culture without the differentiation medium.
Comment: The expression of DSPP protein also increased on day 16 in the co-culture without the differentiation medium. It would be important if the authors could discuss the high DSPP protein expression in the group without the differentiation medium. How did the authors explain the lower DSPP expression associate with higher mineral formation.
Line 277: by co-culturing HERS/ERM cells and human pluripotent stem cell, we have identified human ESCs and iPSCs as practical sources of dental epithelial stem cell differentiation.
Comment: not sure of this statement about dental epithelial stem cell "differentiation".
Reviewer 2 Report
This manuscript describes the obtention of dental epithelial-like stem cells from human ES and iPSC.
The manuscript is well writen albeit it could benefit from a careful proof-reading from a native english speaker. The obtention and characterization of the hES/hiPSC derived cells is clear and convincing.
I have a first concern with the experiments shown on Figure 5. If the cadherin switch and vimentin increase induced by TGFB1 on the HERS/ERM cells used as a positive control is convincing, the author's claim that in the EPI-iPSC, TGFB1 decreases E-CADHERIN and increases N-CADHERIN is not supported by their results. Even if there is maybe a tendancy for E-CADH to decrease, it is not statistically significant. And the so-called increase in N-CADH is almost not visible on the graph and not statistically significant either. In these conditions, the only hint at EMT is from the cells morphology shown in Figure 5a. This is clearly not sufficient to conclude that the EPI-iPSC undergo EMT under TGFB1 like the authors claim it in the results as well as in the discussion sections. The authors should either acknowledge the absence of EMT in their cell line and discuss it or present other experiments showing an EMT.
My second concern is with Figure 6. First, for a reader not familiar with the field of dental stem cells, it is not easy to understand what exactly are hdDPSC compared to HERS/ERM and what kind of differentiation are expected from these different cell types (mesenchymal, epithelial...). A more detailled explanation in the text could help to understand the rationale behing co-culturing hdDPSC with EPI-iPSC and what are the expected results. Second, if as stated the goal of this experiment is to assess the effect of EPI-iPSC cells on odontogenic differentiation, I failed to see how the experimental setup can answer this question. The figure should show the results of markers RT-qPCR, DSPP western-blots and Alizarin red S staining on hdDPSC alone OR with EPI-iPSC (and even EPI-iPSC alone). In the current state, this experiment tells nothing about EPI-iPSC effect on hdDPSC differentiation.
Round 2
Reviewer 1 Report
The authors answered most of the critiques well. However, there are still a few questions concerning this revision.
The authors presented additional data regarding gene expression and mineralization in hdDPSC and hdDPSC+EPI-iPSC groups in Figure A and Figure B. Are these data present in Figure A from day 8 culture or day 20 culture?
From Figure B, hdDPSC only group showed significant but less mineralization compared to those in hdDPSC+EPI-iPSC group. It seems that hdDPSC only group went through odontogenic differentiation.
Thus, it is important to examine DSPP expression in hdDPSC only and hdDPSC+EPI-iPSC groups on day 12, day16 and day 20. It would be good if EPI-iPSC only could be included as a control.
In this revised version line 162, "The expression of ameloblast markers such as Amelogenin, Enamelin, KLK-4 and DSPP was detected when odontogenic differentiation was induced (Figure 6a)".
Comment: The authors included DSPP as an ameloblast marker. It is not appropriate.
Reviewer 2 Report
I thank the authors for their answer to my remarks. The text modifications and additional experiments are nice additions to the manuscript.
Unfortunately I’m still confused by the answer to my last point.
I thought that this study was the first description of the EPI-iPSC cells. So I don’t understand to what previous study the authors are referring and from what study the figures A and B come from.
If it’s unpublished data why not include it in this manuscript ?
Anyway, I consider that a figure for publication should have all appropriate controls even if similar controls were done in different experiments.
So as the authors kindly propose, I think that adding each cell type alone as control is required.